# Bone Marrow Aspirate Matrix: A Convenient Ally in Regenerative Medicine

**DOI:** 10.3390/ijms22052762

**Published:** 2021-03-09

**Authors:** José Fábio Lana, Lucas Furtado da Fonseca, Gabriel Azzini, Gabriel Santos, Marcelo Braga, Alvaro Motta Cardoso Junior, William D. Murrell, Alberto Gobbi, Joseph Purita, Marco Antonio Percope de Andrade

**Affiliations:** 1IOC—Instituto do Osso e da Cartilagem, 1386 Presidente Kennedy Avenue, Indaiatuba 13334-170, Brazil; josefabiolana@gmail.com (J.F.L.); drgabriel.azzini@gmail.com (G.A.); 2Orthopaedic Department—UNIFESP/EPM, 715 Napoleão de Barros St, São Paulo 04024-002, Brazil; ffonsecalu@gmail.com; 3Hospital São Judas Tadeu, 150 Cel. João Notini St, Divinópolis 35500-017, Brazil; marcelobragahsjt@gmail.com; 4Núcleo Avançado de Estudos em Ortopedia e Neurocirurgia, 2144 Ibirapuera Avenue, São Paulo 04028-001, Brazil; contato@hospitalmoriah.com.br; 5Abu Dhabi Knee and Sports Medicine, Healthpoint Hospital, Zayed Sports City, Between Gate 1 and 6, Abu Dhabi 00000 (P. O. Box No. 112308), United Arab Emirates; doctormurrell@gmail.com; 6411th Hospital Center, Bldg 938, Birmingham Ave, Naval Air Station, Jacksonville, FL 32212, USA; 7O.A.S.I. Bioresearch Foundation Gobbi Onlus, 20133 Milano, Italy; gobbi@cartilagedoctor.it; 8Institute of Regenerative Medicine, Boca Raton, FL 33432, USA; jpurita@aol.com; 9Department of Locomotor Apparatus, Federal University of Minas Gerais, Belo Horizonte 31270-901, Brazil; mapa.bhz@terra.com.br

**Keywords:** tissue healing, bone marrow aspirate clot, fibrin matrix, hyaluronic acid, regenerative medicine, orthobiologics

## Abstract

The rise in musculoskeletal disorders has prompted medical experts to devise novel effective alternatives to treat complicated orthopedic conditions. The ever-expanding field of regenerative medicine has allowed researchers to appreciate the therapeutic value of bone marrow-derived biological products, such as the bone marrow aspirate (BMA) clot, a potent orthobiologic which has often been dismissed and regarded as a technical complication. Numerous in vitro and in vivo studies have contributed to the expansion of medical knowledge, revealing optimistic results concerning the application of autologous bone marrow towards various impactful disorders. The bone marrow accommodates a diverse family of cell populations and a rich secretome; therefore, autologous BMA-derived products such as the “BMA Matrix”, may represent a safe and viable approach, able to reduce the costs and some drawbacks linked to the expansion of bone marrow. BMA provides —it eliminates many hurdles associated with its preparation, especially in regards to regulatory compliance. The BMA Matrix represents a suitable alternative, indicated for the enhancement of tissue repair mechanisms by modulating inflammation and acting as a natural biological scaffold as well as a reservoir of cytokines and growth factors that support cell activity. Although promising, more clinical studies are warranted in order to further clarify the efficacy of this strategy.

## 1. Introduction

The rise in musculoskeletal disorders has been a great cause of concern in recent decades. Major health organizations such as the World Health Organization (WHO) confirm that musculoskeletal diseases are the highest contributor to global disability [1]. These health conditions can affect both young and elderly populations by putting bones, joint and muscle tissues at risk and generating a detrimental socioeconomic and psychosocial impact. Current interventional strategies are divided into pharmacological and nonpharmacological alternatives. Popular nonpharmacological strategies usually employ exercise, physical therapy and education, weight loss, physical aids, radiofrequency ablation and—in more severe cases—surgical joint replacement [2]. Pharmacological options only manage pain temporarily and do not target the disease itself [3]. Medical experts may typically prescribe a combination of drugs with the sole objective of blocking inflammatory processes. Nonsteroidal anti-inflammatory drugs (NSAIDs)—e.g., analgesics and steroid drugs—are commonly indicated for this purpose [4]. Application of NSAIDs is particularly concerning. Despite the fact that they effectively alleviate pain, their use is known to be associated with potentially serious dose-dependent gastrointestinal (GI) complications, such as upper GI bleeding [4]. Given this complicated scenario, the rise in demand for novel therapeutic interventions has inspired researchers across the globe to expand the field of regenerative medicine, resulting in the development of orthobiologics. Orthobiologics are products that can rapidly enhance the healing process of orthopedic injuries. These products are naturally found in the body. Currently, some of the most popular biologic products available are platelet-rich plasma (PRP), hyaluronic acid, autologous fat grafts, and bone marrow aspirate/bone marrow aspirate concentrate (BMA/BMAC) [5]. Recent developments have revealed a new variation of bone marrow aspirate-derived products: the BMA clot. This biological product consists of a clot that is naturally formed from the aspirated bone marrow (BM) and contains all of the cellular and molecular BMA components [6]. These components are retained in a fibrin matrix molded by the clot (BMA Matrix), displaying regenerative mechanisms. These mechanisms are relatively similar to fracture hematomas—the highly controlled processes which promote bone remodeling and healing [6]. Regenerative medicine and the associated literature are growing incessantly; however, although BMA-derived products display optimistic results, their mechanisms have not been completely elucidated yet.

This manuscript aims to review the current advances regarding bone marrow aspirate products and proposes the “BMA Matrix” as a feasible part of the orthobiologic landscape in regenerative medicine—one that is being particularly valuable for the treatment of musculoskeletal disorders.

## 2. The Bone Marrow Niche

The bone marrow (BM) is a semisolid tissue located in the central cavities of axial and long bones. This organ contains numerous cellular and molecular components. Typical examples of cell populations residing in the bone marrow can be divided into nonhematopoietic (e.g., pericytes, endothelial cells, osteoblasts, adipocytes and Schwann cells) and hematopoietic (e.g., neutrophils, lymphocytes, megakaryocytes, monocytes and osteoclasts) [7]. All of these cells and associated molecular profiles play important roles in the regulation of the bone marrow niche [8]. In addition to these cell types, there is also the presence of hematopoietic stem cells (HSC) and mesenchymal stem cells (MSC), the two major adult stem cell types in the bone marrow. These adult stem cells are highly praised for their ability to perform self-renewal and differentiation into specific mature cell lineages [9].

Hematopoietic stem cells are comprised of adult progenitor stem cells which are responsible for performing hematopoiesis. In other words, HSCs give rise to all of the blood cells. These cells are present in very small concentrations within the BM; the proportion is approximately 1:10,000 (one HSC for every ten thousand BM mononuclear cells) [10]. The mesenchymal stem cell is the other major adult stem cell type. It is characterized by a set of specific cell surface differentiation markers (CDs). MSCs are known to express a range of cell-lineage specific antigens which differ depending on culture preparation, duration, or plating density [11,12]. Previously, the Mesenchymal and Tissue Stem Cell Committee of the International Society for Cellular Therapy (ISCT) proposed minimal characterization criteria for MSCs, which encompass the following attributes: the cells must be plastic-adherent while in culture; must be able to undergo differentiation in vitro into adipocyte, chondroblast, and osteoblast lineages; must express CD105, CD73 and CD90; must lack expression of CD45, CD11b, CD34 or CD14, CD79α or CD19 and HLA-DR surface markers [13].

MSCs are able to differentiate into mesodermal lineage cells such as cartilage, bone, fat, muscle, meniscus and tendon—a fundamental attribute in regenerative medicine [14]. Additionally, these cells are known for their paracrine effects, being able to manipulate their local microenvironment [15]. These cells are highly advantageous since they do not appear to trigger aggressive immunogenic episodes and can be easily isolated, enabling allogenic transplantation. In such circumstances, these cells should be considered immune-evasive. However, the effects of MSCs in cellular-based therapies tend to be more inclined toward their homing and engraftment abilities into target tissues [16]. It has been hypothesized that MSCs have a rather short life span and are phagocytized by monocytes, subsequently stimulating the production of T-regulatory cells. This sequence of events may very well contribute to overall clinical improvement [17].

Whenever tissue injury occurs, involved cells secrete chemical signals (chemokines) responsible for MSC recruitment. Once recruited into the target tissue, MSCs respond by modulating the wound-healing cascade, reducing apoptosis and fibrosis, attenuating the inflammatory process and stimulating cell proliferation and differentiation via paracrine and autocrine pathways [18]. These effects are due to the ability of MSCs to release key agents, including vasculoendothelial growth factor (VEGF), transforming growth factor beta (TGF-β), stromal-derived factor 1 (SDF-1), and stem cell factor (SCF), among others. Moreover, they can downregulate synthesis of interleukin (IL)-1, IL-6, interferon-γ (IFN-γ) and tumor necrosis factor-α (TNF-α), which are major pro-inflammatory cytokines [19,20,21]. MSCs also display immunomodulatory properties; they are able to inhibit the activation of type 1 macrophages, natural killer (NK) cells, and both B and T lymphocytes [22]. 

Researchers have also zoomed in on interleukin-1 receptor antagonist (IL-1RA), an important cytokine found in bone marrow-derived products. This protein is a competitive antagonist that binds to IL-1B and IL-1a isoforms of cell surface receptors and inhibits IL-1-induced catabolic reactions and inflammatory effects [23]. IL-1Ra has the potential to reduce matrix degradation, as IL-1B reportedly induces MMP-3 and TNF-a gene expression, prostaglandin E2 secretion, chondrocyte apoptosis, and inhibition of collagen deposition [24].

## 3. Bone Marrow Aspirate–Technique and Content

Aspiration of the bone marrow provides a rich source of several cellular and molecular components that elicit numerous effects on tissue regeneration, all of which are particularly beneficial for cartilage, bone and soft tissue injury [25]. This orthobiologic product is usually recommended for the treatment of specific musculoskeletal disorders, e.g., joint arthritis, bone defects, ligament tears, osteonecrosis of the femoral head, nonunion [26]. Bone marrow (BM), and consequently BMA, contains mesenchymal stem cells (MSCs), hematopoietic stem cells (HSCs), endothelial progenitor cells, and other progenitor cells—together with growth factors, including bone morphogenetic proteins (BMP), platelet-derived growth factor (PDGF), transforming growth factor-β (TGF-β), vascular endothelial growth factor (VEGF), interleukin-8 (IL-8), and IL-1 receptor antagonist [27]. 

BMA can typically be harvested from different sites such as the iliac crest, proximal and distal tibia, or calcaneus. However, there is a certain preference to aspirate the bone marrow from the posterior iliac crest, because it has been found that this anatomical region offers a considerable amount of biological material with greater number of osteoblastic progenitors, compared to the tibia and calcaneus [28]. Additionally, different harvesting techniques have been experimented with, generating heterogeneity among reported results in regards to the number of progenitor cells [29]. The aspiration of small volumes of product in different subcortical areas appears to be more efficient, as it yields greater numbers of stem cell progenitors [30]. Medical practitioners may consider the utilization of fluoroscope or ultrasound to guide the aspiration procedure. The use of anatomic landmarks for the average patient and ultrasound guidance for larger individuals may also be recommended. For the average patient, an ideal volume between 90 and 120 milliliters should be drawn, whereas in a large individual, volumes greater than 120–150 mL should be avoided [31,32].

It is important to note that the majority of MSCs lie within subcortical areas, and pericytes tend to be located around the blood vessels (as their name indicates). Therefore, perforating the bone marrow cavity too deeply may not be a wise strategy. It will not yield the greatest amounts of target cells [31]. Figure 1 further illustrates BMA preparation techniques.

In regards to contraindications, patients with significant anemia, systemic infection, active hematologic neoplasm, or individuals who are unable to be anatomically positioned for the procedure should consider other alternatives. If the patient is making use of certain medications, these should be disclosed before the procedure. Examples include: prednisone, since it is known to elicit antianabolic effects; statins, which negatively impact cell proliferation; and nonsteroidal anti-inflammatory drugs, due to their interference in platelet aggregation and clot function [33].

## 4. Coagulated BMA–Potential Applications

The aspirated bone marrow tends to suffer rapid coagulation due to the presence of megakaryocytes, platelets and coagulation factors [34]. Surgeons may circumvent this inevitable event by coating the inside of the syringes with anticoagulant agents, usually acid citrate dextrose solution (ACD) or heparin solutions [35]. Although clot formation may be frowned upon and seen as a technical complication (with good reason), some researchers argue in favor of this outcome. From a biological standpoint, the clot may uncover positive attributes related to the therapeutic value of BMA itself and perhaps further enhance regenerative mechanisms [6]. Thinking critically, adding too many chemicals or exogenous products to the BMA product might interfere in its natural biological potential. Additionally, in order to cause coagulation, platelet degranulation is a required step; this step delivers numerous osteotropic cytokines and growth factors into the site of injury [32]. Another fundamental effect that is often overlooked is the fibrinolytic activity. The fibrinolytic cascade ultimately culminates in fibrin scissions, generating additional factors that are highly beneficial towards cell engraftment [36]. Della Bella et al. [37] illustrated these concepts by attempting to culture BMA in its coagulated and noncoagulated states. Fifteen days post-culture, the researchers observed that the BMA clot culture displayed higher growth kinetics of MSCs in comparison to the samples treated with anticoagulants. Yao and colleagues demonstrated the potential advantages of using bone marrow clots associated with three-dimensional biodegradable scaffolds in promoting extracellular matrix (ECM) scaffold chondrogenic regeneration [38]. It is important to note that due to the abundant presence of fibrin and erythrocytes in clots, there is a certain risk for an initial obstruction of nutrient exchange, which may cause some decrease in the ratio of cell viability. However, it was reported that the same scaffolds led to expressive improvements in cell adhesion, proliferation, and chondrogenic differentiation with porous recanalization in later cultures in comparison to other groups. Overall, there were superior results—in terms of DNA content, expression of Sox9 and RunX2 genes (both highly expressed during chondrocyte and osteoblast proliferation and differentiation), cartilage lacuna-like cells, and ECM accumulation—as well as greater scaffold size and mechanic stability in bone marrow clot cultures. In similar fashion, Pascher et al. conducted an appreciable in vitro study confirming that coagulates formed from BMA may be a useful gene delivery tool for cartilage and possibly other musculoskeletal structures—because it is possible to modify cells within the fluid with an adenoviral vector. Additionally, the clot-derived matrix is completely natural, native to the host, and is the fundamental platform for the healing and repair of mesenchymal tissues [39].

Perhaps one of the most interesting (and sometimes overlooked) features of coagulates is the presence and biological value of fibrin, which is the main component in these hematological manifestations—be it from peripheral blood or bone marrow. The fibrin clot that is formed after an injury can act as a shield to preserve tissues against blood loss and invasion of pathogens and may very well provide a temporary matrix (Figure 1) through which cells can migrate during the repair process [40,41]. Fibrin clots are composed of an agglomeration of platelets embedded in a network of crosslinked fibrin fibers as a result of the cleavage of fibrinogen by serine proteases. This promotes the polymerization of fibrin monomers—a primary event in blood coagulation [42]. This biological structure may also serve as a reservoir of cytokines and growth factors, which are released once activated platelets undergo degranulation [42]. An in vitro study led by Colley et al. [43] concluded that MSCs cultured on fibrin gels display enhanced proliferation potential and are capable of maintaining their osteogenic lineage differentiation potential when compared to MSCs cultured on plastic plates, suggesting that the fibrin matrix could help in maintaining stemness. In further advances, a clinical study [44] revealed optimistic results with the implantation of autologous MSC/fibrin clot constructs in patients suffering from upper limb nonunions. It was argued that the malleable fibrin gel texture in the clot facilitated diffusion of nutrients and differentiation of MSCs toward osteogenic lineage, encouraging tissue integration and healing. This strategy also showed significant safety and effectiveness in the long-term follow up of patients, since there were no occurrences of ectopic neoformation, neoplastic transformation, infection, overgrowth, or refracture. Although complex, researchers believe that the regenerative effects mediated by the clot tend to closely mimic the natural processes that take place in fracture hematomas.

Elaborating further, a BMA clot has superior advantages over other biological products; for instance, the bone marrow fibrin clot appears to offer a significantly higher concentration of VEGF, bFGF, IGF-1, SDF-1, and HGF (which are vital for cell survival, migration, differentiation, and regeneration) in exposed cell cultures than in cells treated with peripheral blood fibrin clot [45]. According to a recent study [46] on a leporine model of long bone defects, the therapeutic potential of autologous BMA clot in the repair of ulnar defects was higher than that of autologous bone grafts, since it was associated with a lower risk of complications. Bone graft fragments (the current main strategy in the use of bone graft augmentation surgeries) may prolong the regenerative cascade, since the body needs to process and resorb microparticles. This is not the case with autologous clot.

Although the therapeutic strategies for the use of BMA clot may vary, most of the studies are intended for the enhancement of cartilage and bone regeneration, being predominantly of preclinical classification. Taking the aforementioned investigations into consideration, the biological potential of bone marrow aspirate products is further illustrated as a beneficial ally in regenerative medicine. 

## 5. Fibrinolytic Mechanisms

It is known that platelets are also found in bone marrow products and carry a milieu of growth factors and proteins (Table 1). These molecules are strongly involved in the fibrinolytic system and may upregulate or downregulate fibrinolytic reactions. The temporal relationship and relative contributions of hematological components (as well as the function of platelets in clot degradation mechanisms) are still a matter of great discussion [47]. The literature presents many investigations thoroughly discussing the role of specific cell populations, proteins, cytokines and platelets, which have long been known for their ability to enhance the healing process. Despite numerous studies, additional hematological components (such as coagulation factors and the fibrinolytic system itself) have also been found to be of equal importance when it comes to wound repair [42]. Fibrinolysis is a complex biological process which relies on the activation of specific enzymes in order to promote degradation of fibrin [42]. Fibrinolytic reactions have been previously discussed by researchers [48], who suggested that fibrin degradation products (FDPs) may actually be responsible for driving tissue repair, preceding the sequence of biological events from fibrin deposition and removal all the way to angiogenesis (which is essential for wound healing).

Regulated by plasmin, the fibrinolytic system exerts key roles in promoting cell migration, growth factor bioavailability, and the regulation of other protease systems involved in tissue inflammation and regeneration [49,50]. The main components present in fibrinolytic reactions, e.g., urokinase plasminogen activator receptor (uPAR) and plasminogen activator inhibitor-1 (PAI-1), are present in MSCs, the specialized cell types necessary for successful wound healing [49].

These cells are known to be mobilized from the bone marrow under adverse conditions (such as severe organ injury) and may therefore be detected in the circulation of individuals with multiple bone fractures, for instance. On the other hand, under specific conditions such as end-stage kidney and liver failure, or during rejection episodes following heart transplantation, these cells may not be detected in blood [51]. It is interesting to note, however, that human bone marrow-derived MSCs may not be detected in the blood of healthy individuals [52]. 

A hypothesis for MSC mobilization has previously been put forward by uPAR, and remains similar to the mobilization process of another cell type present in the bone marrow, the hematopoietic stem cell (HSC). One in vivo study [53] revealed that the urokinase receptor controls mobilization, migration, and differentiation of MSCs, indicating a supportive role for the fibrinolytic system in regenerative processes. Vallabhaneni et al. observed that uPAR was required for MSC mobilization from the BM of mice who were stimulated with granulocyte colony-stimulating factor (G-CSF), as an insignificant amount of these cells was mobilized in uPAR deficient mice when compared to the eight-fold increase in wild types [53]. Furthermore, uPAR-deficient mice demonstrate upregulated expression of G-CSF and stromal cell-derived factor 1 (CXCR4) receptors in BM; the downregulation of uPAR inhibits migration of MSCs, including the ones of human origin. Lastly, the authors concluded that either down- or upregulation of the urokinase receptor can inhibit or stimulate MSC differentiation, respectively [53]. Others have shown that the glycosylphosphatidylinositol-anchored uPA receptor can control adhesion, migration, proliferation, and differentiation events. This can be achieved through the activation of certain intracellular signaling pathways, such as the prosurvival phosphatidylinositol4,5-bisphosphate 3-kinase/Akt and ERK1/2 signaling, as well as the focal adhesion kinase (FAK) [54,55].

## 6. The Role of Hyaluronic Acid in the BMA Matrix

Hyaluronic acid (HA) or hyaluronan is an anionic, nonsulfated glycosaminoglycan that is widely distributed throughout connective, epithelial, and neural tissue [56]. Hyaluronan is also an important component of articular cartilage in which it exists as a coating surrounding each chondrocyte. It carries lubricant properties for tendons and joints, and reduces the breakdown of extracellular matrix by inhibiting metalloproteinase synthesis. Its anti-inflammatory properties downregulate the activity of tumor necrosis factor-alpha and interleukin-1, two major inflammatory aggressors.

In a respectable study, authors evaluated the use of fibrin and hyaluronic acid mixtures as an ally in the regenerative potential of bone marrow concentrates [57]. Histologically, the bone marrow concentrate and hyaluronic acid plus fibrin groups exhibited significant chondral regeneration, with faster and more complete healing of the osteochondral hole in the trochlear region of rabbit knees (in comparison to the hyaluronic acid plus fibrin only group and controls). Furthermore, scanning electron microscopy images of hyaluronic acid/fibrin composites mixed with bone marrow cells revealed that mixing fibrin with hyaluronic acid enabled it to polymerize from solid to porous structures through which cells attached and could be stimulated to proliferate.

Hyaluronic acid and fibrin mixtures are supposed to have a fundamental role in creating an effective connection between cells (Figure 1). In this sense, HA might influence the polymerization process of fibrin turning it into a more spongy structure that would otherwise be formed by the solely natural set of fibrinogen and thrombin. This arrangement seems to be critical for cell adherence, both locally and peripherally.

From a safety standpoint, fibrin sealants are biological adhesives that mimic the final step of the coagulation cascade; therefore, an association with hyaluronic acid injection may be considered a biologically safe and natural approach [58].

In a gout model study, articular cartilage regeneration with autologous bone marrow aspirate and hyaluronic acid was demonstrated in detail [59]. Interestingly, full-thickness cartilage defects were created and animals were given either no injections or weekly sessions of HA or HA plus BMA for three weeks. The addition of HA alone managed to improve the quality of repaired tissue (e.g., hyaline-like cartilage). However, the combination of HA plus BMA yielded the best results with vertical orientation of chondrocyte nests, and the presence of type II collagen and proteoglycans in the intermediate and deep cartilage layer (more safranin O staining than the other two groups); and type I collagen confined to the superficial layer and perichondrium. Chondrocyte hypertrophy was not observed. Authors concluded that after arthroscopic subchondral drilling, the technique may optimize articular cartilage repair.

In a randomized controlled trial, articular cartilage regeneration with autologous peripheral blood stem cells (PBSC) combined with HA versus HA alone was studied [60]. Fifty patients with ICRS (International Cartilage Repair Society) grade 3 and 4 lesions of the joint were allocated to either receive a series of weekly injections either HA or PBSC + HA. Subjective IKDC (International Knee Documentation Committee) scores and MRI (magnetic resonance imaging) scans were obtained preoperatively and postoperatively at serial visits. Additionally, a second-look arthroscopy and biopsy were performed to grade the regenerated tissue through ICRS II (International Cartilage Repair Society Visual Assessment Scale II). On ICRS II histologic and morphologic MRI scores, the combined injections yielded statistically better results reflecting the quality of cartilage repair, which was represented by the sustained clinical improvement at 24 months in the PBSC + HA group.

Previous contributions have been made, in which the effect of HA mixed with PRP (the PRP matrix) showed better clinical outcomes (Western Ontario and McMaster Universities Arthritis Index (WOMAC) and Visual Analogue Scale (VAS)) over HA or PRP alone [61]. The authors concluded that the lubrication and support to the extracellular matrix that HA provided seemed to enable earlier functional benefits to the PRP injection. This may ameliorate rehabilitation and anticipate a patient’s return to routine activities. Considering these observations, we argue that the BMA Matrix may share similar mechanisms and effects—which have been previously suggested in the numerous studies cited in this manuscript.

## 7. Authors’ Note

BMA Matrix could be enhanced when combined with low doses of recombinant human growth hormone (GH), glutathione and low doses of dexamethasone through the use of a three-way stopcock to connect and mix syringes before injections (Figure 2). Preclinical data have shown that coinjection of HA and GH was more effective in modifying structures and symptoms when compared to the injections of HA alone in a leporine model of osteoarthritis (OA) [62]. GH is known to be able to stimulate cell growth, reproduction and regeneration. It stimulates insulin-like growth factor 1 (IGF1) production by the liver. IGF1 in turn elicits growth effects on almost every cell in the body, especially skeletal muscle, cartilage and bone cells [63]. Research has revealed that morphoangiogenesis (neocapillary blood vessels that penetrate into the bone layer) found in the knee injected with recombinant human GH may play a role in cartilage regeneration [64]. GH also aids in overcoming inflammatory pain through the indirect role of cortisol on prostaglandins [65].

Experiments with glutathione and its precursor molecule, N-acetylcysteine (NAC), have shown interesting results in the literature for the management of musculoskeletal disorders. They have demonstrated efficacy in reducing inflammation markers as well as significant improvements in pain and functional outcomes, due to their protection against oxidative stress and overproduction of reactive oxygen species [2].

Dexamethasone, in its turn, has recently been shown to rescue cartilage matrix loss and chondrocyte viability in animal studies and cartilage explant models of tissue injury and post-traumatic osteoarthritis. This indicates a possible role for dexamethasone as a disease-modifying drug for a limited amount of time, if used at low doses [66]. In this sense, under arthritic stress, dexamethasone could prevent proteolysis in the cartilage matrix and might reduce IL-17-induced nitric oxide (NO) synthesis, inhibiting the production of IL-6 and inducible nitric oxide synthase (iNOS) transcription. This suggests a potential pathway for the anti-inflammatory effects of dexamethasone in cartilage tissue [67].

## 8. Conclusions

Given that bone marrow contains a complex environment of many cell populations, the use of BMA clot may represent a viable alternative, able to reduce the costs and drawbacks linked to the expansion of cells from the BM, in terms of regulatory compliance and the investment required for the manipulation and storage of biological materials. To elaborate, obtaining BMAC is invasive, requires closed systems during preparation, and the positive results are strongly correlated with the number of stem cells [68].

Furthermore, due to the high number of variables that exists when processing other orthobiologics (such as PRP and BMAC), BMA carries the advantage of not requiring additional steps and protocols for its preparation. The BMA Matrix represents another alternative to enhance the mechanism by which the clot acts on the housing of transplanted cells and recruitment of the circulating ones. Although we have presented numerous benefits regarding the application of BMA-derived products, additional studies are highly warranted to further improve our understanding of the novel orthobiologics and the enhancement of regenerative processes in tissue injury.

## Figures and Tables

**Figure 1 ijms-22-02762-f001:**
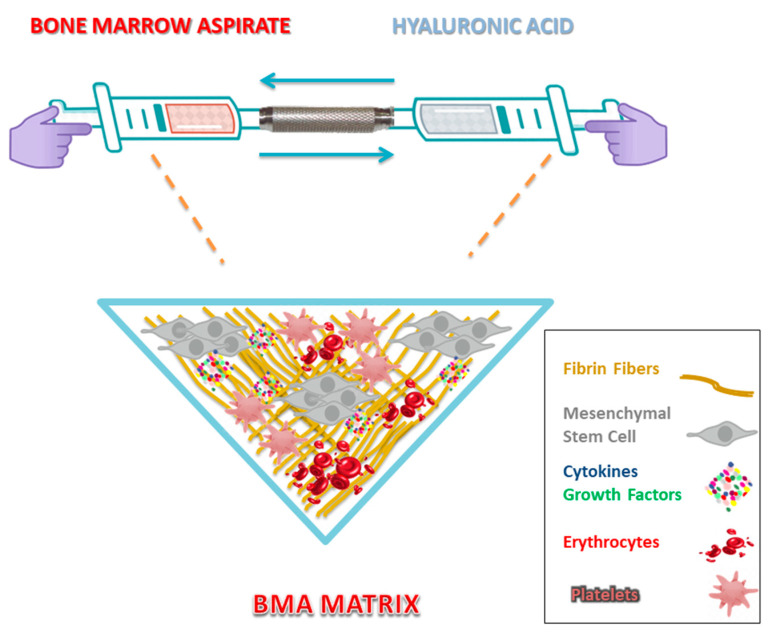
Bone marrow aspirate (BMA) Matrix. The combination of BMA with hyaluronic acid (HA) enables the formation of the BMA matrix, proposing a novel strategy towards the enhancement of regenerative mechanisms and possible synergistic effects. The crosslinked fibrin fibers derived from the BMA clot form a natural scaffold, supporting cellular activity and reconstruction of damaged tissue.

**Figure 2 ijms-22-02762-f002:**
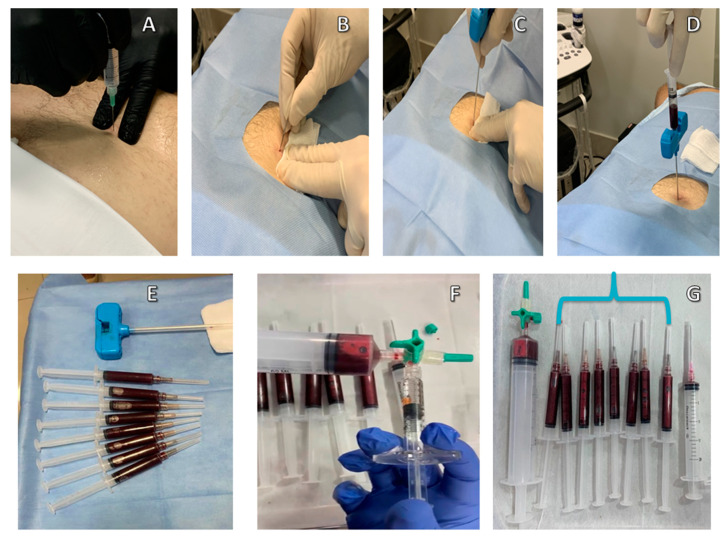
Preparation of fresh autologous bone marrow aspirate matrix. (**A**) Application of local anesthesia; (**B**) target point incision; (**C**) posterior iliac crest puncture; (**D**) aspiration of fresh bone marrow; (**E**) anticoagulant-free syringes filled with bone marrow aspirate; (**F**) bone marrow aspirate mixed with hyaluronic acid; (**G**) Bone Marrow Aspirate Matrix inside the 3 mL syringes (indicated by the blue bracket).

**Table 1 ijms-22-02762-t001:** The biological role of growth factors present in platelet α granules.

Name	Abbreviation	Biological Role
Platelet-derived growth factor	PDGF	Increases expression of collagen, proliferation of bone cells, fibroblast chemotaxis and proliferative activity, macrophage activation.
Vascular endothelial growth factor	VEGF	Triggers angiogenesis, chemotaxis of macrophages and neutrophils, migration and mitosis of endothelial cells, and increases permeability of blood vessels.
Fibroblast growth factor	FGF	Promotes the growth, proliferation and differentiation of chondrocytes and osteoblasts, and stimulates proliferation of mesenchymal cells.
Transforming growth factor-β	TGF-β	Augments production of collagen type 1, stimulates angionesis and chemotaxis of immune cells, inhibits osteoclast formation and bone resorption
Hepatocyte growth factor	HGF	Secreted by mesenchymal cells, HGF stimulates mitogenesis, cell motility, and matrix invasion.
Fibroblast growth factor	FGF	Regulates cellular proliferation, survival, migration, and differentiation.
Epidermal growth factor	EGF	Stimulates proliferation and differentiation of epithelial cells, promotes secretion of cytokines by mesenchymal and epithelial cells.
Insulin-like growth factor	IGF	Promotes cell growth and differentiation, stimulates collagen synthesis and recruits cells from bone, endothelium, epithelium and other tissues.
Insulin-like growth factor-1	IGF-1	By eliciting anabolic effects, this hormone plays a key role in cellular growth.

## Data Availability

Not applicable.

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
