# Peer review of "Bone Marrow Aspirate Matrix: A Convenient Ally in Regenerative Medicine"

_ijms, 2021, doi:10.3390/ijms22052762_

Round 1

Reviewer 1 Report

This submitted paper reads more like a newspaper article than a true medical Journal submission. This begins with the title which includes "A convenient ally In Regenerative Medicine". One would wonder what exactly a "convenient ally" actually means. This terminology is used throughout this manuscript. The paper includes some interesting conjecture regarding the use of coagulated bone marrow aspirate and how this may provide additional blood products for tissue healing along with a mixture of bone marrow aspirate with how ironic acid. The exact type of how ironic acid is not described. The paper ends with anecdotal additional information which includes combining both hormone and low-dose dexamethasone along with glutathione. All this is then mixed to create a cocktail that could provide additional benefits. Obviously, this is all conjecture without supportive data for this mixture. A paper that focuses on potential application of the coagulated BMA may be considered if the authors desired to better focus what has been submitted. Clearly, more appropriate scientific language should be used as well.

Reviewer 2 Report

This is an interesting and very well presented overview of the potential of autologous bone marrow aspirate derived biologic products in regenerative medicine for the treatment of complicated orthopedic conditions. The text is clear and illustrations are good. My main questions are concerning the biologic understanding, and translational pathways necessary for preclinical to clinical applications. 

These include:

  • As the authors are no doubt aware, ex-vivo changes occur in the BM aspirate during the clotting process, in addition to those associated with haemostasis. For example, cytokine production continues, or may be upregulated during this stage with, for example IL-8 production particularly affected. What steps should be taken to 'fix' the aspirated material in an as near in vivo state as possible, or are these changes in fact advantageous as they may lead to a more efficacious product for certain applications?
  • In the contest of the above point, what advantages or disadvantages would anti-coagulated BM aspirate present? 
  • What steps are required for processing the BM aspirate clot to a form, or forms, suitable for re-infusion? In other words, how do you proceed from BM aspiration to re-infusion of the desired product? A figure may be a good way to illustrate this.
  • What are the in vitro storage conditions for the product(s), and how are the products administered.
  • As the authors are no doubt aware, homeostasis requires a very fine balance between activation of the coagulation cascade, inhibitors and fibrinolysis. What are the effects of using BM aspirate clot derived products on this balance? Is there any evidence that systemic imbalance may occur, even with localised administration?
  • Again, as the authors are no doubt aware, the use of stem cells from PB, after appropriate mobilisation, has largely replaced BM aspirate in autologous stem cell transplants in treatment of malignancies, in particularly those of haematological origin. Do they envisage the possibility that this route may be adequate for some of their proposed applications, or is the BM matrix a completely necessary requirement?
  • I'm not sure I agree with the overstated difficulty of the high number of variables that exists when processing other orthobiologics such as platelet-rich plasma (PRP). Perhaps that is not the best example as it is a well standardised, routine procedure in clinical practice. 
  • What ethical approvals are required/already obtained for such applications.

The authors should address these questions and perhaps provide another figure to map the preclinical to clinal pathway. This would give the reader (and this reviewer) a better understanding of requirements for realistic clinical application, and outline the necessary questions to be addressed, and practical obstacles to be resolved, to enable a very interesting hypothesis to be delivered in the clinic.  

Reviewer 3 Report

A very interesting article on an important topic, especially given the fact that with an increase in the life expectancy of modern society, diseases of the musculoskeletal system come out on top of the causes of premature disability. Currently, a significant part of healthcare is engaged in solving difficult problems of treating diseases of the musculoskeletal system with high resource technological and economic costs. The authors concentrate on a specific aspect of cartilage repair using a coagulated bone marrow transplant. The components and mechanisms of the formation of bone marrow aspirate (BMA) clots are described in this review. Authors highlight the various benefits of technology and additional perspective ingredients (hyaluronic acid, growth factors, and hormones) that enhance cartilage repair. However, some paragraphs and sentences are difficult to understand, both due to the lack of basic information and due to the rather cryptic and sometimes incorrect language. The figures and tables are clear and helpful. In general, I find the review useful and interesting for the reader. Below are the step-by-step suggestions for the authors to improve the text:

Major comments.

  1. There is no information about the authors' affiliation. Please provide in accordance with the IJMS-MDPI requirements for review manuscripts (Title, Author list, Affiliations, Abstract, Keywords) (see IJMS-MDPI Author's Guide).
  2. The list of references is not properly annotated. Please, use the IJMS-MDPI Instructions for Authors guide to compile a list of references.
  3. The abstract exceeds the word count recommended in the author's guide for IJMS-MDPI. https://www.mdpi.com/journal/ijms/instructions#preparation
  4. Lines 14-17: The sentence needs to be edited to clarify the meaning.
  5. Lines 18-21: The sentence is cryptic, please explain the meaning.
  6. Line 25: What authors would like to say be by this conclusion? “Matrix may still shine through and offer a glimmer of hope for certain healthcare challenges”. This statement is not specific and does not clarify what was covered in this review or what healthcare challenges and hopes were identified by the authors.
  7. Line 25: What did the authors want to say in this conclusion? " Matrix may still shine through and offer a glimmer of hope for certain healthcare challenges " This statement is not specific and does not clarify what was considered in this review. What problems, suggestions, and challenges in cartilage repair did the authors identify?
  8. In the abstract, please provide a more structured summary of your research, eg problems in this area, a summary of the content of the review, why this review is necessary and important; clear short conclusions. See journal guidelines: https://www.mdpi.com/journal/ijms/instructions#preparation For more information see https://libguides.eastern.edu/c.php?g=116063&p=756459
  9. Of course, no one can forbid you to directly repeat the main text of the conclusions in the abstract. However, the abstract assumes that you briefly report your findings, and at the end of the manuscript give a detailed explanation of your findings. Unfortunately, in the abstract, the authors used vague, general phrases instead of specifically reflecting the essence of their review, and in the conclusions in the main text, they did not reveal the main aspects that they discussed in the review.
  10. Line 180: The authors rightly note that a large number of erythrocytes are present in the thrombus. After the formation of a clot, lysis of erythrocytes is observed, which causes the release of toxic substances. This process is a negative factor for the use of BMA for the regeneration of cartilage and bone tissue. The authors did not discuss this aspect of the proposed method.

Unclear paragraphs or sentences

  1. Line 144-146
  2. Line 154-158
  3. Line 188-190: It is unclear about the delivery of the gene to cartilage.
  4. Line 235-237: What molecules do the authors mean. Are they "platelets" or "these factors"? What are these factors?
  5. Line 265-268: Does G-CSF contribute to the disruption of migration, or does uPAR deficiency contribute to this disruption of migration?
  6. Line 364: Conclusions: " some drawbacks linked to BM concentration and expansion" It is not clear what the authors mean by these " drawbacks ", please explain this phrase/sentence in the conclusions and edit or delete this vague sentence in the abstract.
  7. Line 369-371: Please clarify the sentence and explain what kind of " anecdotal experience on cell recruitment " the authors had as they wrote this review.

Minor comments.

  1. Line 11: Please pay attention to the reduction of some sentences, removing unnecessary introductory words and vague wording. This is especially recommended in the abstract. g “Over the course of the past two decades”-> Recently,
  2. Line 31-33: Please simplify the sentence.
  3. Line 33-35: Please clarify the sentence.
  4. Line 50: Word “substances” is not exactly cover complex tissue definition like cells, stromal matrixes, soluble factors, and mineral compounds.
  5. Line 62-64: Since regenerative medicine is a very wide area of research, please specify the area of application of BMA-matrix.
  6. Line 126: “non-union” please, clarify the word.
  7. Line 168: “interfere (in or with) its natural biological potential”
  8. Lines 282, 283 “Bone marrow concentrates”, maybe you mean bone marrow aggregates? What is the difference between “bone marrow aggregates” and “bone marrow concentrates”?
  9. Line 313: Please explain the abbreviation “IKDC scores and MRI scans”
  10. Line 334: excessive bracket “Application of local anesthesia (; “
  11. Line 347 and Figure 2; “G) Bone Marrow Aspirate Matrix”. It is unclear on panel G where exactly the BMA matrix is displayed. Please indicate this with an arrow (or using other signs).

Round 2

Reviewer 2 Report

I thank the authors for their detailed responses. I am still unclear about several factors, such as how widely this approach is applied, what the evidence base is (eg whether there have been many controlled trials, or the feasibility of same), and what the timelines are between BM aspiration and application. Also, has any attempt been made to quantify the key components by, for example, measuring cytokines etc in centrifuged BM clot supernatant. However, the paper is well presented and referenced, and will be of interest as a discussion paper.

Reviewer 3 Report

Dear authors, all questions should be addressed and answered in the text of the manuscript. I am sorry that during editing I have uploaded one version of the manuscript, which does not match your more advanced version. That is why your line numbering has been shifted. Please see below for those comments that were not answered.

The list of references is not properly annotated: E.g please see reference #17 and #39 (…. et. al) it is not in format for MDPI citation.  All names of the authors have to be present.

No affiliation has yet been added to the manuscript. Please provide in accordance with the IJMS-MDPI requirements (see IJMS-MDPI Author's Guide). https://www.mdpi.com/journal/ijms/instructions#preparation

I am citing for you from the guideline: “The PubMed/MEDLINE standard format is used for affiliations: complete address information including city, zip code, state/province, and country. At least one author should be designated as corresponding author, and his or her email address and other details should be included at the end of the affiliation section. Please read the criteria to qualify for authorship. After acceptance, no updates to author names or affiliations will be permitted”.

Please explain the meaning below. The sentence is not entirely clear to the readers. Please change the text of the manuscript: “Drugs and other substances that should also be disclosed before the procedure include but are not limited to: prednisone, since it is known to elicit anti-anabolic effects; statins, which negatively impact cell proliferation; and non-steroidal anti-inflammatory drugs due to their interference on platelet aggregation and clot function

Line 82-84: Please work on this proposal to further clarify and make it readable.  “as a feasible orthobiology in regenerative medicine” ->  e.g “ This manuscript aims to review the current advances regarding bone marrow aspirate products and  proposes the “BMA Matrix” as a feasible part of the orthobiologic landscape in regenerative medicine being particularly valuable for musculoskeletal disorders treatment.”

Round 3

Reviewer 3 Report

Thank you for your work in correcting the manuscript. I hope it will be accepted for publication soon

Good luck